# Expression of Calcitonin Gene-Related Peptide and Calcitonin Receptor-like Receptor in Colorectal Adenocarcinoma

**DOI:** 10.3390/ijms25084461

**Published:** 2024-04-18

**Authors:** Robert-Emmanuel Șerban, Mioara-Desdemona Stepan, Dan Nicolae Florescu, Mihail-Virgil Boldeanu, Mirela-Marinela Florescu, Mircea-Sebastian Șerbănescu, Mihaela Ionescu, Liliana Streba, Nicoleta-Alice-Marinela Drăgoescu, Pavel Christopher, Vasile-Cosmin Obleagă, Cristian Constantin, Cristin Constantin Vere

**Affiliations:** 1Department of Gastroenterology, University of Medicine and Pharmacy of Craiova, 200349 Craiova, Romania; drrobert.serban03@gmail.com (R.-E.Ș.); cristin.vere@umfcv.ro (C.C.V.); 2Research Center of Gastroenterology and Hepatology, University of Medicine and Pharmacy of Craiova, 200638 Craiova, Romania; 3Department of Infant Care-Pediatrics-Neonatology, University of Medicine and Pharmacy of Craiova, 200349 Craiova, Romania; desdemona.stepan@umfcv.ro; 4Department of Immunology, University of Medicine and Pharmacy of Craiova, 200349 Craiova, Romania; 5Department of Pathology, University of Medicine and Pharmacy of Craiova, 200349 Craiova, Romania; mirela.florescu@umfcv.ro; 6Department of Medical Informatics and Biostatistics, University of Medicine and Pharmacy of Craiova, 200349 Craiova, Romania; mircea.serbanescu@umfcv.ro (M.-S.Ș.); mihaela.ionescu@umfcv.ro (M.I.); 7Department of Oncology, University of Medicine and Pharmacy of Craiova, 2 Petru Rares Str, 200349 Craiova, Romania; liliana.streba@umfcv.ro; 8Department of Anesthesiology and Intensive Care, University of Medicine and Pharmacy of Craiova, 200349 Craiova, Romania; alice.dragoescu@umfcv.ro; 9Department 5, “Carol Davila” University of Medicine and Pharmacy, 050447 Bucharest, Romania; christopher.pavel@gmail.com; 10Department of Surgery, University of Medicine and Pharmacy of Craiova, 200349 Craiova, Romania; cosmin.obleaga@umfcv.ro; 11Department of Radiology and Medical Imaging, University of Medicine and Pharmacy of Craiova, 200349 Craiova, Romania; cristian.constantin@umfcv.ro

**Keywords:** calcitonin gene-related peptide, calcitonin receptor-like receptor, colorectal adenocarcinoma

## Abstract

Colorectal cancer is one of the most widespread types of cancer that still causes many deaths worldwide. The development of new diagnostic and prognostic markers, as well as new therapeutic methods, is necessary. The calcitonin gene-related peptide (CGRP) neuropeptide alongside its receptor calcitonin receptor-like receptor (CRLR) could represent future biomarkers and a potential therapeutic target. Increased levels of CGRP have been demonstrated in thyroid, prostate, lung, and breast cancers and may also have a role in colorectal cancer. At the tumor level, it acts through different mechanisms, such as the angiogenesis, migration, and proliferation of tumor cells. The aim of this study was to measure the level of CGRP in colorectal cancer patients’ serum by enzyme-linked immunosorbent assay (ELISA) and determine the level of CGRP and CRLR at the tumor level after histopathological (HP) and immunohistochemical (IHC) analysis, and then to correlate them with the TNM stage and with different tumoral characteristics. A total of 54 patients with newly diagnosed colorectal adenocarcinoma were evaluated. We showed that serum levels of CGRP, as well as CGRP and CRLR tumor level expression, correlate with the TNM stage, with local tumor extension, the presence of lymph node metastasis, and distant metastasis, and also with the tumor differentiation degree. CGRP is present in colorectal cancer from the incipient TNM stage, with levels increasing with the stage, and can be used as a diagnostic and prognostic marker and may also represent a potentially new therapeutic target.

## 1. Introduction

Colorectal cancer represents one of the biggest health problems, being the third type of cancer in terms of case numbers worldwide and causing approximately 1 million deaths annually [1,2]. Although there are numerous screening programs, the number of patients diagnosed in later stages is still high [3,4]. There is a need for the development of new markers in the early detection of colorectal cancer and also new therapeutic methods.

Calcitonin gene-related peptide (CGRP) is a neuropeptide consisting of 37 amino acids, which, in addition to its vasodilator effect, is considered a multifunctional regulatory agent [5,6,7]. Besides central and peripheral nervous tissue, it is widely distributed in connective tissue and actively participates in the process of developing and modeling various structures [8]. CGRP is expressed in the gastrointestinal tract in numerous diseases such as chronic inflammations, ulcers, polyps, and adenocarcinomas, in a different way depending on the location of the lesions [9,10,11].

In the literature, there are very little data related to the expression of CGRP in colon carcinomas (CRCs) and the existing information is relatively old and contradictory. Thus, although there were investigations that did not support the regulatory role of neuropeptides on the growth of cancer cells [12,13], some studies indicated the role in the growth and metastatic dissemination of numerous tumors [8,14].

Calcitonin receptor-like receptor (CRLR), together with receptor-associated modifying protein (RAMP 1), forms a receptor complex for calcitonin gene-related peptide (CGRP) [15,16,17]. CRLR is also a receptor for adrenomedullin when it bonds with RAMP 2 and 3 [5,17,18]. Unlike the receptors of other neuropeptides, the effects of CRLR and RAMP 1 are less studied, although their expression is documented in malignant cell lines of the prostate and mammary gland [19,20].

Although the relationship between neuropeptides and cancer is known, a study to establish the exact relationship between CGRP and CRC has not been realized, considering the fact that it could represent a useful diagnostic and prognostic marker. In this study, the aim was to evaluate the calcitonin gene-related peptide level in colorectal cancer patients’ serum and, together with calcitonin receptor-like receptor, to determine their tumoral levels and toidentify their potential role in the pathogenesis of colorectal adenocarcinomas. We also wanted to evaluate the diagnostic and prognostic role and the correlation of CGRP with different clinicopathological characteristics in CRC patients. We also made comparisons between the presence and level of CGRP and CRLR at the tumor level, observing whether they correlate with each other and determining whether they could represent possible therapeutic targets.

## 2. Results

This study included 54 newly diagnosed patients with colorectal cancer aged between 43 and 87 years, with an average age of 69.4 ± 9.9, including 40 men and 14 women, and a control group of 18 patients with similar age and gender distribution (13 men and 5 women). The most frequent location of the primary tumor was in the left colon with 26 cases, followed by the right colon and then the rectum with 19 and 9 cases, respectively.

### 2.1. Serum CGRP in Colorectal Patients’ Characteristics

The mean values of serum CGRP are indicated in Table 1, presented by patients and tumor characteristics.

The comparison of CGRP serum levels in different TNM stages showed significant variations (F = 7.6, *p* < 0.001). Serum CGRP had high levels in patients with CRC, starting from TNM stage I, compared to the control group (mean diff. −5.33, *p* < 0.001), with increased levels with stage advancement. This means that higher levels correlate with more advanced disease. Tukey’s post hoc analysis revealed that the biggest mean difference was between TNM stage I and II (mean diff −1.24, *p* = 0.068), which shows us that once it has outgrown muscularis propria, the tumor starts to secrete an increasing amount of CGRP. The smallest mean difference was found between stage II and III (mean diff. 0.61, *p* = 0.54), showing that there are no significant differences when the tumor exceeds muscularis propria and in the presence of lymph node metastasis.

In terms of the size and extension of the primary tumor (T-stage), there are also significant variations (F = 7.5, *p* < 0.001). The smallest mean differences are between T1 and T2 (mean diff.—0.95, *p* = 0.62), proving that the tumor increasingly secretes larger amounts of CGRP after it exceeds muscularis propria; the biggest mean difference is seen between T3 and T4 (mean diff. −1.147, *p* = 0.19), showing that CGRP is present in very large amounts in tumors that invade surrounding organs.

Regarding lymph node metastasis, there are also significant differences among group means (F = 6.3, *p* = 0.003). CGRP had higher levels in patients with lymph node metastasis compared to patients without lymph node metastasis (N0 vs. N1 mean diff. −1.53, *p* = 0.003; N0 vs. N2 mean diff −1.06, *p* = 0.19). Serum CGRP had a slight decrease in patients with N2 compared to N1 (mean diff 0.47, *p* = 0.75), showing that it can be correlated with the presence of lymph node metastasis, but not with the number of involved lymph nodes.

CGRP had higher levels in patients presenting distant metastases (mean diff. 1.4, *p* = 0.007).

Regarding tumor grading, there is also a significant difference between the subgroups’ mean values (F = 32, *p* < 0.001). There is a very small mean difference between G2 and G3 (mean diff. −0.19, *p* < 0.0090), but a bigger mean difference between G1 and G2 (mean diff. −1.9 *p* < 0.001), showing that its dosage can differentiate between well-differentiated tumors and other types, but cannot differentiate well between moderately differentiated and poorly differentiated tumors.

Regarding the patients’ characteristics, serum CGRP had higher levels in men compared with women (mean diff. 0.25, *p* = 0.61) and in elderly patients in comparison with younger patients (mean diff. 0.29, *p* = 0.5), but the differences between groups were not statistically significant.

The level of serum CGRP was defined as low or high, based on the CGRP mean value, and the patients were divided into two subgroups according to this level. The Kaplan–Meier curve showed a survival at 20 months of 19.6 months for patients with a low level and 18.2 for patients with a high level of serum CGRP, which means a poorer prognosis for the patients with a high level of serum CGRP (log rank test, χ^2^ = 1.91, *p* = 0.16) (Figure 1).

We can, therefore, observe that CGRP is at high levels in older patients and male patients, in TNM stages III and IV, in more advanced tumors (T3–T4), in the presence of lymph node metastasis, regardless of the number of lymph nodes involved. The presence of distant metastases (M1) and those with moderately and poorly differentiated tumors (G2–G3) were also associated with a lower survival time, showing us that it can be used as a good diagnostic and negative prognostic marker in colorectal adenocarcinoma.

### 2.2. Histopathological and Immunohistochemical Tumor Analysis: CGRP and CRLR Immunohistochemical Identification and Final Staining Score

In this study, the following tumor characteristics prevailed: non-mucinous CRCs (77.8%), low-grade (66.7%), pT3 tumor extension (53.7%), without regional nodal metastases (53.7%) or distant metastases (79.6%), and tumors in stages IIA and I (51.8%) (Table 2). The investigated CRCs presented metastases in the regional lymph nodes (pN1–2) in 46.3% of cases: pN1a (2 cases—3.7%), pN1b (8 cases—14.8%), pN2a (11 cases—20.4), and pN2b (4 cases—7.4%), respectively. CRCs presented distant metastases (pM1) in 20.4% of cases: pM1a (7 cases—13%) and pM1c (4 cases—7.4%), respectively. At the level of the invasion site, tumor budding (Bd1–3) was present in 46.3% of cases and poorly differentiated clusters (PDC) in 20.4% of cases; at the same time, vascular (VI) and perineural (PI) invasion were identified in 48.1% and 31.5% of cases (Table 2). For all characteristics except tumor subtype and tumor budding, the differences in CGRP FSS expression were statistically significant between categories. CRLR FSS only presented statistically significant differences for tumor grade, poorly differentiated clusters, tumor extension (pT), regional lymph node metastasis (pN), distant metastasis (pM), and tumor stage (Table 2).

There were no statistically significant differences in CGRP FSS and CRLR FSS between genders or age groups (Table 3).

CRLR and CGRP immunoreactions were identified in all analyzed cases at the cytoplasmic level, both in the CRC parenchyma and stroma and in the adjacent normal mucosa. The number of marked stromal elements in the adjacent normal mucosa was lower than in the tumor stromal areas, and the markings of normal epithelial cells were of weak intensity, especially at the basal pole, in a maximum of 5–10% of the cells.

#### 2.2.1. CGRP Histopathological and Immunohistochemical Characteristics

In the case of CGRP, the reactions were present in the tumor stroma, in rare lymphocytes, fibroblasts, macrophages and leukocytes, apoptotic cells, and enteroendocrine, but especially in the associated plasma cells, which showed intense markings (Figure 2a). The reactions were identified in the cytoplasm of tumor cells and apoptotic cells. For the analyzed tumor group, the mean number of CGRP-labeled cells was 52.5 ± 19, with a variable intensity of reactions and a mean FSS of 5.5.

An analysis of CGRP reactions indicated significantly higher differences in high-grade CRC, with PDC in front of invasion, with advanced tumor extension (pT) and nodal (pN) and distant metastases (pM), which were in advanced stages (Table 2). The differences of CGRP FSS were statistically significant for TNM stages, tumor grading, and CRC with poorly differentiated clusters at the limit of statistical significance for CRC with Bd in the invasion front and with perineural invasion, and with insignificance in relation to tumor type and vascular invasion (Table 2).

We see that in the case of CGRP, the FSS is higher in patients with TNM III and IV, tumors invading adjacent organs (T4), metastases in more than three lymph nodes (N2), patients with distant metastases (M1), tumors with low differentiation grades (G3), a tumor budding score of 3, the presence of poorly differentiated clusters (PDC 1), the presence of vascular invasion (VI 1) and perineural invasion (PI 1), and with small higher difference in FSS in the case of mucinous tumors. Thus, CGRP is associated with more advanced disease, more advanced tumors, more lymph node metastases, the presence of distant metastases, and also with the histopathological and immunohistochemical characteristics of aggressiveness and invasiveness, and less with tumor type (mucinous/non-mucinous). This shows us the potential role of tumor and disease progression and invasiveness of CGRP in the case of colorectal adenocarcinomas, regardless of tumor type (mucinous or non-mucinous).

#### 2.2.2. CRLR Histopathological and Immunohistochemical Characteristics

At the level of the CRC stroma, CRLR reactions were identified in eosinophils, fibroblasts, some lymphocytes, macrophages, endothelial cells, and muscle fibers of medium caliber vessels, but also at the level of adipose tissue; the markers were also present in inflammatory intraepithelial mononuclear elements and enteroendocrine cells (Figure 3).

Reactions were present in the tumor parenchyma, including in cells undergoing apoptosis. For the entire analyzed group, the average number of marked tumor cells was 53.7 ± 16.8, the intensity of the reactions was variable, and the average FSS value was 5.9. A semiquantitative analysis of CRLR indicated a significantly higher mean FSS in high-grade CRC with PDC at the front of invasion, with advanced tumor extension (pT), and with nodal (pN) and distant metastases (pM), which were found in advanced stages (Table 2, Figure 3b–d). At the same time, the CRLR FSS differences were statistically significant for TNM stages, tumor grade, and CRC with poorly differentiated clusters, at the limit of statistical significance in the case of CRC with vascular invasion and insignificance in relation to the tumor type, the presence of Bd in the invasion front, and perineural invasion (Table 2).

For CRLR, FSS was higher in patients with TNM stages III and IV, in T3 and T4 tumors, in patients with lymph node metastases (N1–2), distant metastases (M1), low differentiation grade tumors (G3), a tumor budding score of 3, the presence of poorly differentiated clusters (PDC 1), and the presence of vascular invasion (PI 1). CRLR also had an increased staining score, both in the presence or absence of perineural invasion, and in both tumor subtypes (mucinous/non-mucinous). This shows us, as in the case of CGRP, the potential role of CRLR in the progression and invasiveness of colorectal adenocarcinomas, but that, unlike CGRP, the CRLR staining score cannot differentiate between tumors with perineural invasion and between mucinous/non-mucinous tumors.

### 2.3. Comparison between the Immunohistochemical Final Staining Score of CGRP and CRLR according to Patients’ Clinical Data and Tumor Characteristics

The final staining score of both CGRP and CRLR was divided into two categories: low FSS (1–4) and high FSS (6–12). We observed that CGRP had more patients with low FSS than CRLR (29 vs. 23), and CRLR had more patients with high FSS than CGRP (31 vs. 25). From Table 2 and Table 3, comparing the average levels of FSS, we observed that both CGRP and CRLR have higher levels in older patients and in men, with CRLR having higher average levels than CGRP.

Regarding the tumor characteristics, both had increasing levels with the advancement in TNM stage, with CRLR having higher levels in stages I and IV and CGRP in stage II; stage III had the same mean levels. Regarding the tumor extension (T), both had increasing levels with the more advanced tumors. CGRP only had higher average levels in T1 tumors, while for T2–T4 tumors, CRLR had higher levels. Both CGRP and CRLR had approximately equal average levels in patients without lymph node metastasis, with increasing levels as more lymph nodes were invaded, and with CRLR having higher average levels than CGRP in stages N1 and N2. Also, in patients without distant metastasis, the average levels were equal, but in patients with M1, CRLR had higher average levels than CGRP. Regarding tumor pathological grade, patients with low-grade tumors (G1–G2) had similar mean levels of CGRP and CRLR. In patients with high-grade tumors (G3), CRLR had higher mean levels.

We can conclude that, regarding the average levels of FSS, CRLR has higher levels than CGRP in both men and women, as well as for older patients. Also, in early tumor stages and tumor grading, both have similar levels, but in advanced stages (TNM IV), locally advanced tumors (T3–T4), invasion of lymph nodes (N1–2), distant metastases (M1), and more undifferentiated tumors (G3), CRLR has higher average FSS levels than CGRP.

We divided the FSS of CGRP into low and high levels, and, according to the clinicopathological characteristics of patients with CRC, we observed that the increased levels of CGRP are significantly associated with age (χ^2^ = 4.7, *p* = 0.03), TNM stage (χ^2^ = 6.686, *p* = 0.042), and tumor pathological grade (χ^2^ = 7.7, *p* = 0.021) at the statistical limit of significance for tumor extension (χ^2^ = 5.534, *p* = 0.052) and are not significantly associated with age, lymph node, or distant metastasis (Table 4).

We also divided the FSS of CRLR into low and high levels, and, according to the clinicopathological characteristics of patients with metastasis, observed that the increased levels of CRLR are significantly associated with gender (χ^2^ = 7.001, *p* = 0.008), age (χ^2^ = 637, *p* = 0.046), and lymph node metastasis (χ^2^ = 0.012, *p* = 0.04) and are not significantly associated with TNM stage, tumor extension, distant metastasis, or tumor pathological grade (Table 5).

The Kaplan–Meier survival analysis showed that the average survival time at 20 months for patients with low CGRP FSS was 19.5 months and for those with high CGRP FSS it was 18.4 months (log rank test, χ^2^ = 1.092, *p* = 0.29) (Figure 4a); CRLR had similar survival time: 19.7 months for low CRLR FSS and 18.5 months for high CRLR FSS (log rank test, χ^2^ = 4.578, *p* = 0.032) (Figure 4b).

There is a statistically significant strong positive correlation between CGRP and CRLR staining score for CRC patients (r = 0.613, *p* < 0.001, Pearson test). This shows us their synergistic role in colorectal cancer, and the fact that the neuropeptide–receptor binding has a role in increasing invasiveness and metastasis at the tumor level.

## 3. Discussion

In human tissues, there are two forms of CGRP (α/β), and α-CGRP is the most studied [5,21,22]. There are data in the literature that indicate a particular interaction between nerve structures and tumor cells that promotes tumor progression. The interaction between tumor cells and nerve fibers supports the progression of the disease, as demonstrated in pancreatic and prostate cancer, where tumor cells that are near the nerves have less apoptosis and a higher expression of Ki-67 in comparison to tumor cells some distance from the nerve cells. Also, certain cells have neuropeptide receptors on their surface, increasing communication between nerves and tumor cells, as demonstrated in esophageal cancer, when the inhibition of neuropeptide receptors decreases cell proliferation [23]. Nerve fibers communicate with tumor cells through neuropeptides, such as CGRP or NGF (nerve growth factor), through which they achieve important processes in tumor progression such as angiogenesis and metastasis [24,25,26].

A direct effect of CGRP on tumor growth is suggested, especially through binding to the RAMP 1 component of the receptor; in any case, the CGRP-RAMP1-CRLR complex seems to have a direct effect on tumor cells. By promoting processes such as angiogenesis and lymphangiogenesis, perineural invasion, and modulating inflammation through pro-inflammatory cytokines, CGRP plays an important role in tumor growth and progression [19,27,28,29,30].

CGRP is involved in the motility stimulation (increase by 30–40%), migration, and invasion of tumor cells, together with other neuropeptides such as adrenomedullin and calcitonin, and seems to cooperate within the metastasis process [20,31,32]; the process seems to be related to or a component of the epithelial–mesenchymal transition, which is involved in the progression of carcinomas at different locations [33,34]. CGRP stimulates tumor migration and invasion through the mitogen-activated protein kinase (MAPK) signaling pathways. In animal models, the inhibition of bone metastases of prostatic adenocarcinoma was found when CGRP antagonists were administered, which designates this neuropeptide as a potential therapeutic target [14].

Numerous studies have indicated increased serum CGRP levels in breast, lung, thyroid, and prostate cancers [14,35,36,37,38,39]. However, the data in the literature related to the tissue expression of CGRP in the normal and tumor colon are practically non-existent.

By stimulating tumor angiogenesis/lymphangiogenesis, the proliferation and migration of endothelial cells, and the vasodilator effect, CGRP participates in the growth and survival of tumor cells, and the administration of antagonists reduces the expression of VEGF and the formation of new vessels [8,20,29,35,40]. In our study, we found the expression of CGRP in medium-caliber vascular endothelium.

The increased expression of CGRP is generally associated with high-grade carcinomas [13,37]. In our study, CGRP immunoexpression was identified in all CRC cases and was associated with high grade, advanced stage, the presence of PDC and Bd in the invasion’s front, and perineural invasion, an aspect that is difficult to compare with data from the literature, which generally refers to other locations of malignant tumor processes. Some studies have indicated that, although CGRP has less of an effect on tumor proliferation, the neuropeptide exerts its growth-stimulating effects through mechanisms involving other cells such as connective stromal cells and macrophages, which indicates a paracrine mechanism [14]. The process is supported by the stimulatory role of CGRP in the healing processes of connective tissue, including bone defects in association with internal or external osteoforming factors [41]. Also, over the past two decades, some authors have suggested a possible autocrine tumor stimulation activity involving neuropeptides [42,43]. These mechanisms are also suggested by the reactions obtained in our study, which were present in frequent stromal and inflammatory cells.

CRLR is a constitutive G protein that forms receptor heterodimers for CGRP if it becomes connected with RAMP 1. The receptor complex is connected with two cytoplasmic proteins [5]. In the study by Wende B et al., regardless of the human tissue investigated, CRLR was expressed in endothelial cells, immune cells (monocytes, T lymphocytes, macrophages), and vascular smooth muscle; some aspects of this were described in this study, with the particularity that the markings at the level of plasma cells were intense. There are studies carried out at the level of the gastrointestinal tract that indicated an increased expression of CRLR in the tumor areas compared to the adjacent non-tumorous tissue, and the expression of the receptor was associated with high tumor grade and the advanced stage of the tumor [44,45]. In our study, the increased expression of CRLR was additionally associated with the presence of PDC in the invasion front and with vascular invasion. Also, in the study by Wende B et al., CRLR was present in 50% of colon carcinomas; however, the massive immunoexpression of CRLR was identified in all normal gastrointestinal specimens analyzed, at the mucosa level, of the endocrine and immune cellular elements [20]. The markings were present in all analyzed CRCs, but also in the non-tumor areas, with lower markings. CRLR was observed in normal and tumor apoptotic cells. Some data indicate the involvement of neuropeptides in the survival of tumor cells by activating the MAPK/ERK pathway and autocrine/paracrine stimulation of Bcl-2 expression [46].

In this article, we observed the presence of CGRP in the patient’s serum at a much higher level than in the control group patients. It correlates both with the tumor stage and with the pathological tumor differentiation grade, with increased levels also being correlated with lower survival. Also, with the help of immunopathological staining, we observed that at the tumor level, the increased expression of both CGRP and CRLR is associated with advanced tumor stages, with undifferentiated tumors, with a high score of tumor budding, with the presence of poorly differentiated clusters, and with perineural and vascular invasion. Considering the association of increased levels of CGRP and CRLR with tumors and advanced tumor characteristics and their correlation, we can say that the CGRP/CRLR combination has a potential role in tumor invasiveness and disease progression and does not represent an effect of advanced disease. This means that they can represent therapeutic targets.

Considering that the CRC patients had higher serum CGRP levels than the control group, and considering the association between high levels of CGRP in CRC patients’ serum and advanced stages and decreased survival rate, we can say that it can be used both as a diagnostic and prognostic marker in patients with colorectal cancer.

In our study, we showed that serum CGRP is present from the early stages of colorectal cancer, with increasing levels as we advance through the TNM stages and also with higher levels in the more undifferentiated tumors (tumor grade). This means that it can represent a diagnostic and a prognostic marker. It correlates with both CGRP and CRLR at the tumor level, in terms of clinical data—gender and age—and also in terms of tumor and disease characteristics. Serum CGRP correlates with the TNM stage, with higher levels in more advanced stages and also with tumor differentiation grade: lower levels in G1 tumors, and higher levels in G2 and G3 tumors, but without significant differences between the last two. Also, increasing levels correlate with more invasive tumors (T). Higher levels also correlate with distant metastasis (M) and the presence of lymph node metastasis (N), but not with the number of lymph node metastases.

The comparisons between the FSS of CGRP and CRLR show us that there is a strong correlation between them and also that increased levels are associated with advanced stages of the disease, which suggests that the neuropeptide–receptor link is associated with tumorigenesis processes.

Currently, there are therapies targeted toward CGRP and its receptor in the prevention of migraine—Erenumab—and considering the association between CGRP and colorectal cancer, they may represent an effective treatment in this type of cancer, but numerous studies and trials with large, heterogeneous patients’ groups must be performed [47,48].

The dosage of CGRP in the patient’s serum is not specific for colorectal cancer, but considering the data shown in our article, its levels can suggest the presence of this type of cancer and can guide the investigations for the diagnosis. It can also suggest the colorectal cancer patient’s prognosis. The presence of CGRP and CRLR at the tumor level has been demonstrated in other types of cancer, so we cannot say that it is specific for colorectal cancer, but this could lead to comparative studies between different types of cancer, to be able to determine for which one it is more specific.

### Limitations

The polyclonality of one of the antibodies used for IHC reactions can be considered a cause of the presence of non-specific markers or the accentuated background of the reactions. To reduce these effects and the lack of availability of an antihuman CGRP monoclonal antibody for IHC on paraffin-embedded tissues, we performed standardization, the use of reaction controls, the blocking of non-specific sites, and the double and consensual quantification of markers by two experienced pathologists. Another potential limit, which would have brought some information related to the more precise tissue localization of CGRP and CRLR, is related to the possibility of performing double immunohistochemical or fluorescent reactions to indicate the presence of possible colocalizations; this aspect is important to establish the tissue sources of the proteins and the autocrine and paracrine mechanisms involved in tumor stimulation and that can be carried out in future studies. However, due to the limited information in the literature related to the expression of the two proteins in the case of CRC, we believe that, first of all, the separate characterization of the immunoexpression of the markers at the level of the parenchyma and the tumor stroma had to be carried out in order to establish a quantitative and descriptive immune profile as completely as possible. In this context, studies are needed to investigate not only descriptively, but also quantitatively, the significance of CRLR and CGRP reactions in inflammatory cells from the tumor stroma and from the level of the invasion front, and also from the level of the adjacent mucosa, especially since some studies indicate the association of postoperative complications for CRC with the composition and intensity of the inflammatory response [49].

The group of patients was homogeneous and from in a single center, and the follow-up period was relatively short at only 20 months. An extended multicentric study on heterogeneous groups of patients must be performed for safer conclusions.

## 4. Materials and Methods

This study initially included 75 patients with newly diagnosed colorectal cancer, of which 54 met the inclusion criteria, and a control group consisting of 18 healthy patients, with similar ages and female/male ratios. The study was previously approved by the Ethics Committee of the University of Medicine and Pharmacy of Craiova, no. 4/21 January 2022. The diagnosis and reporting of CRC were carried out according to the latest criteria developed by the WHO working group for tumors of the digestive system [50]. The patients were followed for 20 months from the diagnostic time.

### 4.1. Patient Selection

Inclusion criteria: informed consent signed by the patients; patients without a personal history of cancer; patients who have not received chemotherapy, radiotherapy, immunotherapy, or biological therapies; patients without autoimmune diseases and important liver and kidney diseases. The patients were diagnosed at the Craiova County Emergency Clinical Hospital and at the Gastroenterology and Hepatology Center at the University of Medicine and Pharmacy Craiova. We took the patients’ medical history, completed a clinical exam, and then the patients underwent a colonoscopy with biopsy and a complete set of blood tests. The extent of the disease was evaluated by CT/MRI scan and then the patients underwent surgical intervention or pre-surgical oncological treatment, depending on the disease stage. After the surgical intervention, the postoperative fragments were sent for histopathological and immunohistochemical examination.

Other patients’ characteristics and clinical data were also taken into account, such as environment (urban or rural), lifestyle (obesity, smoking, heavy alcohol consumption), level of education (primary school, high school, higher education), and family history (colorectal cancer, other digestive cancers, other types of cancers).

### 4.2. Enzyme-Linked Immunosorbent Assay (ELISA)

The analysis of CGRP in the patient’s serum was performed by the enzyme-linked immunosorbent assay (ELISA) test, using the CGRP ELISA kit from Elabscience (Houston, TX, USA) according to the protocol. Venous blood (5 mL) was collected from the patients, which was centrifuged for 10 min at 3000× *g*. Then, the serum was kept at lower temperatures (between −20 and −80 degrees Celsius), keeping it until the moment when the reagent was applied.

Colorectal cancer patients were divided according to the mean value of serum CGRP (7.26 ng/mL) into 2 categories: those with low levels of CGRP and those with high levels of CGRP.

### 4.3. Histopathological (HP) and Immunohistochemical (IHC) Assays

The biological material for the histopathological (HP) and immunohistochemical (IHC) study was represented by colectomy surgical specimens, which were fixed in 10% formalin, processed according to the histopathological standard technique, and stained with hematoxylin–eosin. The inclusion criteria in the HP/IHC study were represented by the diagnosis of CRC and antigenic reactivity to Vimentin of the optimal sections selected as being representative of the investigated case.

We considered as optimal sections those who presented with most of the HP parameters of aggressiveness, which were represented by tumor type and grade, tumor budding (Bd), the presence of undifferentiated clusters at the tumor front (PDC), vascular invasion (VI), perineural invasion (PI), and tumor stage (pTNM). For the homogeneity of HP/IHC results, CRCs were considered conventional (non-mucinous) and mucinous; the non-mucinous group included both conventional carcinomas without other specifications (NOS) that did not present components of other subtypes, as well as carcinomas that presented micropapillary, serous, or other components with signet ring cells, which, in this study, represented no more than 25% of the respective tumor areas. In this study, vascular invasion referred to both lymphatic and blood vessel invasion; although, theoretically, the two types of vessels can be differentiated based on the morphology of the wall and the luminal blood elements (red cells, lymphocytes), the appearance can be difficult to interpret, especially in cases of small caliber blood vessels.

The grading of the mucinous CRC subtype is still considered problematic in the literature, but follows the same grading rules according to the presence of the glandular component [51]. For grading, we used the two-step system, with well/moderately differentiated tumors (G1/G2) being considered low-grade and poorly differentiated (G3) tumors, high-grade; also, for tumoral Bd, we used the three-level system: low (0–4 buds), intermediate (5–9 buds), and high (≥10 buds) [50].

Serial sections of 3 µm were made for IHC analysis, which were deparaffinized (xylene, 3 × 5 min), hydrated (alcohol solutions with decreasing concentrations of 100%, 90%, and 70%, ×5 min each), and treated with hydrogen peroxide (0.3%, 15 min) for blocking endogenous peroxidase and with bovine serum albumin (0.01% in phosphate-buffered saline) for blocking non-specific sites. Antigenic recovery was carried out according to the manufacturer’s instructions in citrate solution pH 6. In this study, the primary polyclonal antibody antihuman calcitonin gene-related peptide (CGRP, Abcam, Cambridge, UK) and the primary monoclonal antibody antihuman calcitonin receptor-like receptor (hCRLR, R&D Systems, Minneapolis, MN, USA) were used in dilutions of 1/75. The system used to visualize the reactions was represented by the EnVision™ FLEX+ System (code K8002, Dako, Santa Clara, CA, USA), and the development was carried out with the chromogen DAB (3,3′-diaminobenzidine tetrahydrochloride). To validate the reactions, negative (omitting the primary antibody) and positive external controls (prostatic acinar adenocarcinoma for CGRP with positivity in tumor cells and lung tissue for hCRLR with positivity in vascular endothelium) were used. After counterstaining with hematoxylin, the sections were dehydrated (alcohol solutions with decreasing concentrations of 70%, 90%, and 100%, ×5 min each), clarified (xylene, 3 × 5 min), and permanently mounted.

The IHC reactions were descriptively analyzed at the tumor level and the adjacent mucosa, highlighting the particular aspects of the epithelial and stromal markers. Also, the reactions were quantitatively quantified at the level of the tumor parenchyma by two experienced pathologists who, in case of inconsistencies, repeated the quantification until a consensus was established. For each case, by evaluating 10 microscopic fields (MFs) ×400, a final staining score (FSS) was established, which took into account the number of marked cells and the intensity of the reactions. The final FSS for each case was represented by the product of the scores of the number of labeled cells and the intensity. The number of marked cells for each MF was represented by the percentage of marked tumor cells, finally establishing an average for each case and in case of intensity, the score that prevailed was taken into account. Thus, for the number of positive cells, the used scores were 1 (5–25%), 2 (26–50%), 3 (51–75%), and 4 (>75%), while the scores for the intensity of the reactions were 1 (weak), 2 (moderate), and 3 (high).

The patients were divided according to the final staining score into low FSS (1–4) and high FSS (5–12).

### 4.4. Statistical Analysis

For the statistical analysis, the scores were considered low for values of 1–4 and high for 6–12. The reactions were considered positive if the value of the percentage of marked cells was at least 5%. The images were acquired using a Motic Panthera DL microscope (Motic China Group Co., Xiamen, China), equipped with Motic Images Plus 3.0 ML software.

The statistical analysis followed the prevalence of the lesions within the HP categories that were analyzed and between the HP parameters and the associated FSS scores. For each category, we expressed continuous variables as means for the number of marked cells, as well as median values for FSS. To assess the normality of continuous data, we used the Shapiro–Wilk test. To compare the FSS associated with the HP parameters and the numerical values of the marks obtained, we used comparison tests represented by the Chi-square (χ^2^), Student’s *t* test, and ANOVA for group comparisons, in the case of normally distributed variables, otherwise Mann–Whitney U or Kruskal–Wallis H tests were used. Post hoc analysis was performed using Tukey’s multiple comparison tests.

We used MdCalc (https://www.mdcalc.com/), Easymedstat (https://www.easymedstat.com/), and SPSS 10 (Statistical Package for the Social Sciences) software applications for statistical analysis and comparisons. The results were considered significant for *p* < 0.05 values and at the limit of statistical significance for *p* ≤ 0.09 values.

## 5. Conclusions

Calcitonin gene-related peptide has higher levels in colorectal adenocarcinoma patients’ serum and can be used as a diagnostic and prognostic marker. Also, CRLR and CGRP immunoexpression is widespread in CRC and is associated with reserved prognostic histological criteria of the lesions. The simultaneous presence of reactions in numerous inflammatory cells suggests the existence of autocrine and paracrine mechanisms for stimulating tumor progression. The markers used in this study may be useful for identifying aggressive CRC and for stratifying patients for antineoplastic therapy. Future studies are needed to establish the potential therapeutic targets for CRLR and CGRP in colorectal cancer.

## Figures and Tables

**Figure 1 ijms-25-04461-f001:**
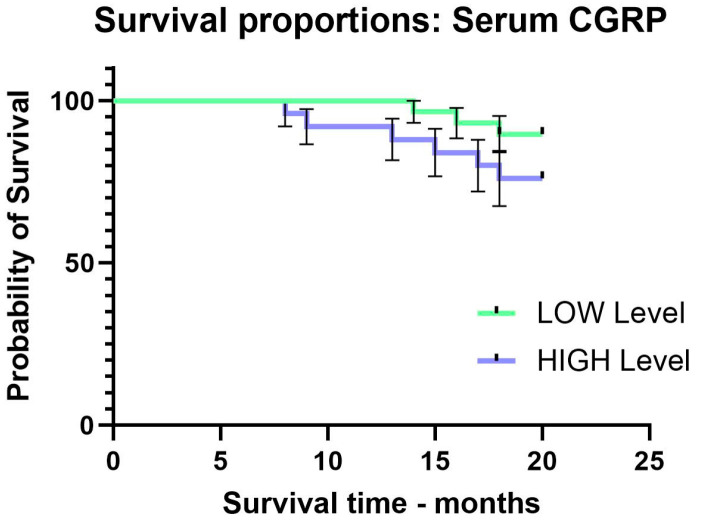
Survival time for CCR patients according to serum CGRP levels (low or high).

**Figure 2 ijms-25-04461-f002:**
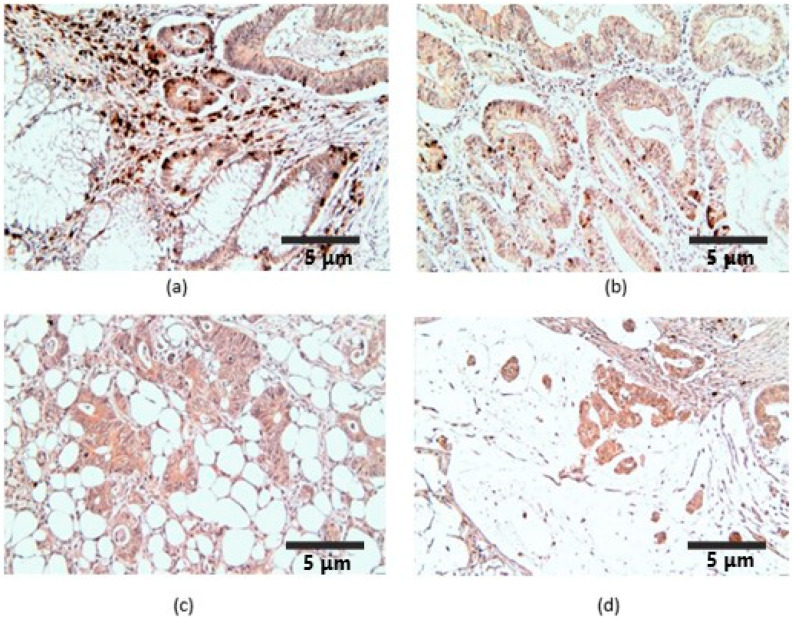
CGRP immunoexpression, ×200. (**a**) CRC and adjacent nontumoral mucosae; (**b**) non-mucinous low-grade CRC; (**c**) non-mucinous high-grade CRC; (**d**) mucinous CRC.

**Figure 3 ijms-25-04461-f003:**
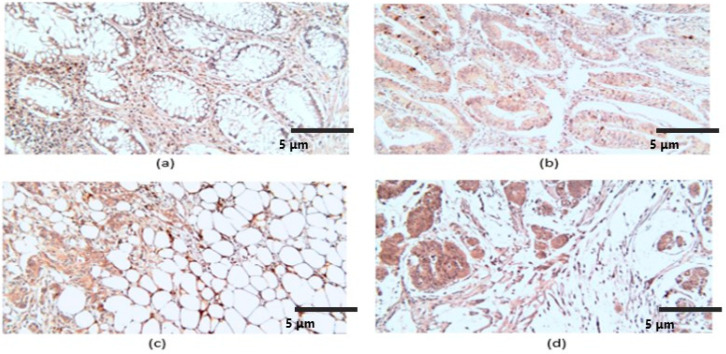
CRLR immunoexpression, ×200. (**a**) Adjacent nontumoral mucosae; (**b**) non-mucinous low-grade CRC; (**c**) non-mucinous high-grade CRC; (**d**) mucinous CRC.

**Figure 4 ijms-25-04461-f004:**
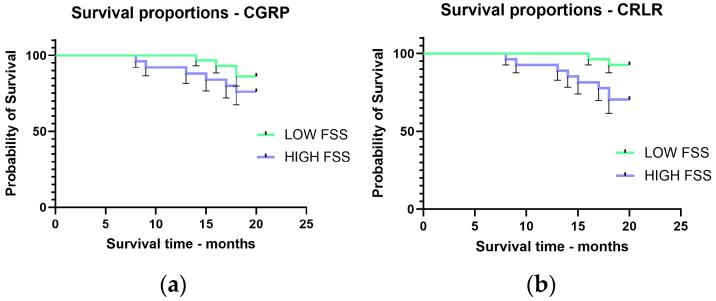
Survival time for CCR patients according to serum FSS levels (low or high): CGRP (**a**), CRLR (**b**).

**Table 1 ijms-25-04461-t001:** Serum CGRP mean levels according to colorectal cancer patients and tumor characteristics.

Patient and Tumor Characteristics		No. Cases (%)	Serum CGRP Mean (ng/mL)	*p*-Value
Gender	Male	40 (74.1)	7.4	0.61 *
	Female	14 (25.9)	6.9	
Age	≥69 years old	30 (55.6)	7.5	0.5 *
	<69 years old	24 (44.4)	7	
Tumor Grade (G)	G1	10 (18.6)	5.7	<0.001 **
	G2	26 (48.1)	7.6	
	G3	18 (33.3)	7.7	
Tumor extension (pT)	T1	4 (7.4)	5.6	<0.001 **
	T2	11 (20.4)	6.7	
	T3	29 (53.7)	7.5	
	T4	10 (18.5)	8.8	
Regional lymph node metastasis (pN)	N0	29 (53.7)	6.6	0.003 **
	N1	10 (18.5)	8.4	
	N2	15 (27.8)	7.9	
Distant metastasis (pM)	M0	43 (79.6)	7	0.007 *
	M1a,c	11 (20.4)	8.4	
TNM stage	Control Group	18	0.5	<0.001 **
	I	12 (22.2)	5.9	
	II	16 (29.6)	7.1	
	III	15 (27.8)	7.8	
	IV	11 (20.4)	8.4	

* Student’s *t* test. ** ANOVA test.

**Table 2 ijms-25-04461-t002:** Tumor characteristics’ case distribution, median FSS, and statistical meaning. FSS: final staining score (median levels).

Tumor Characteristics		No. Cases (%)	CGRP FSS (Median)	*p*-Value	CRLR FSS (Median)	*p*-Value
Tumor subtype	Non-mucinous	42 (77.8)	4.0	0.467 *	6.0	0.537 *
	Mucinous	12 (22.2)	5.0		6.0	
Tumor grade	Low grade (G1–G2)	36 (66.7)	4.0	0.001 *	4.0	<0.0005 *
	High grade (G3)	18 (33.3)	9.0		8.5	
Tumor budding (Bd)	Bd 0 (absent)	29 (53.7)	4.0	0.081 **	4.0	0.054 **
	Bd 1	11 (20.4)	4.0		6.0	
	Bd 2	9 (16.7)	6.0		6.0	
	Bd 3	5 (9.2)	9.0		9.0	
Poorly differentiated clusters (PDC)	PDC 0 (absent)	43 (79.6)	4.0	0.015 *	4.0	0.001 *
	PDC 1 (present)	11 (20.4)	9.0		9.0	
Vascular invasion	VI 0 (absent)	28 (51.9)	4.5	0.033 *	4.0	0.065 *
	VI 1 (present)	26 (48.1)	6.0		6.0	
Perineural invasion	PI 0 (absent)	37 (68.5)	4.0	0.013 *	6.0	0.188 *
	PI 1 (present)	17 (31.5)	9.0		6.0	
Tumor extension (pT)	T1	4 (7.4)	1.5	<0.0005 **	1.5	<0.0005 **
	T2	11 (20.4)	3.0		4.0	
	T3	29 (53.7)	4.0		6.0	
	T4a,b	10 (18.5)	8.5		9.0	
Regional lymph node metastasis (pN)	N0	29 (53.7)	4.0	0.001 **	4.0	<0.0005 **
	N1a,b	10 (18.5)	5.0		6.0	
	N2a,b	15 (27.8)	9.0		8.5	
Distant metastasis (pM)	M0	43 (79.6)	4.0	0.007 *	5.0	<0.0005 *
	M1a,c	11 (20.4)	6.0		10.0	
Tumor stage	I	12 (22.2)	2.0	0.001 **	2.0	<0.0005 **
	II	16 (29.6)	4.0		4.0	
	III	15 (27.8)	6.0		6.0	
	IV	11 (20.4)	7.0		10.0	

* Mann–Whitney U test. ** Kruskal–Wallis test.

**Table 3 ijms-25-04461-t003:** Patient characteristics’ case distribution, median FSS, and statistical meaning. FSS: final staining score (mean levels).

Patients’ Characteristics		No. Cases (%)	CGRP FSS (Median)	*p*-Value *	CRLR FSS (Median)	*p*-Value *
Gender	Male	40 (74.1)	6.0	0.776	6.0	0.642
	Female	14 (25.9)	2.0		3.0	
Age	≥69 years old	30 (55.6)	4.0	0.531	4.0	0.130
	<69 years old	24 (44.4)	5.0		6.0	

* Mann–Whitney U test.

**Table 4 ijms-25-04461-t004:** Correlation between CGRP final staining score and clinicopathological parameters of CCR patients according to low and high levels of FSS.

Parameter		CGRP FSS Low	CGRP FSS High	χ^2^ Test	*p*-Value
Gender	Male	12	18	5.098	0.239
	Female	17	7		
Age	≥69 years old	18	22	4.700	0.030
	<69 years old	11	3		
TNM stage	I	10	2	6.686	0.042
	II	8	8		
	III	6	9		
	IV	4	7		
Tumor extension (pT)	T1 + T2	10	2	5.534	0.052
	T3	14	16		
	T4	5	7		
Regional lymph node metastasis (pN)	N0	18	13	0.5567	0.455
	N1 + N2	11	12		
Distant metastasis (pM)	M0	24	19	0.378	0.538
	M1	5	6		
Tumor grade	G1	10	1	7.700	0.021
	G2	14	18		
	G3	5	6		

**Table 5 ijms-25-04461-t005:** Correlation between CRLR final staining score and clinicopathological parameters of CCR patients according to low and high level of FSS.

Parameter		CRLR FSS Low	CRLR FSS High	χ^2^ Test	*p*-Value
Gender	Male	8	22	7.001	0.008
	Female	15	9		
Age	≥69 years old	14	26	3.637	0.046
	<69 years old	9	5		
TNM stage	I	8	4	4.434	0.218
	II	7	9		
	III	5	10		
	IV	3	8		
Tumor extension (pT)	T1 + T2	8	4	3.695	0.157
	T3	11	19		
	T4	4	8		
Regional lymph node metastasis (pN)	N0	13	18	0.012	0.040
	N1 + N2	10	13		
Distant metastasis (pM)	M0	19	24	0.219	0.639
	M1	4	7		
Tumor grade	G1	8	3	5.143	0.763
	G2	11	21		
	G3	4	7		

## Data Availability

Data are contained within the article.

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
