# Peer review of "Expression of Calcitonin Gene-Related Peptide and Calcitonin Receptor-like Receptor in Colorectal Adenocarcinoma"

_ijms, 2024, doi:10.3390/ijms25084461_

Round 1

Reviewer 1 Report

Comments and Suggestions for Authors

The manuscript aims to show the correlation between CGRP and CRLR expression with CRC stage, or the tumor progress with serum tests from CRC and health patients. 

There are several flaws that makes the manuscript must be improved:

1. The CGRP with CRLP are neuropeptides that have been widely studied and found highly related to pain transmission and immune response. These are two biological processes level up with tumor progress. So it is not convincing enough to say they are suitable staging marker for CRC. Besides, the relationship between neuropeptides and cancer have been well studied including for CRC around 2020 so the novelty can also be improved.

2. The manuscript is in a descriptive manner without necessary conclusive and interpretive paragraphs. Such as for the last paragraph of introduction. It would be better to have one paragraph to summarize the experiments with relative results and interpretive conclusions containing biological meaning.

3. The statistical tests and presents can be improved. The example can be the missing significance test for CGRP expression between different tumor stages. The differences need to be present with statistical calculation. 

Author Response

Response to Reviewer 1 Comments

Thank you for taking the time to review our manuscript.

All changes in the article are highlighted in red

The manuscript aims to show the correlation between CGRP and CRLR expression with CRC stage, or the tumor progress with serum tests from CRC and health patients. There are several flaws that makes the manuscript must be improved:

Point 1: The CGRP with CRLP are neuropeptides that have been widely studied and found highly related to pain transmission and immune response. These are two biological processes level up with tumor progress. So it is not convincing enough to say they are suitable staging marker for CRC. Besides, the relationship between neuropeptides and cancer have been well studied including for CRC around 2020 so the novelty can also be improved.

Response 1: Although neuropeptides have been studied in cancer, including colorectal cancer, there are no studies that focuses directly on the presence and relationship between CGRP and CRLR in colorectal cancer. We believe that through the mechanisms of influencing tumorigenesis already studied (like angiogenesis), the studied molecules would rather have a causal role, and aren’t an effect in cancer, including colorectal cancer. Also, the novelty would be the fact that showing the association both at the serum level and at the tumor level of CGRP with the presence of increased levels in patients with CRC compared to healthy patients in the control group and also the association with advanced stages and low survival we demonstrated that they can be used in the diagnosis and especially in the prognosis of colorectal cancer patients, and may represent possible therapeutic targets.

Point 2: The manuscript is in a descriptive manner without necessary conclusive and interpretive paragraphs. Such as for the last paragraph of introduction. It would be better to have one paragraph to summarize the experiments with relative results and interpretive conclusions containing biological meaning.

Response 2: We added concluding sentences and interpretive paragraphs, in introduction section (the last paragraph) and in discussions section, in paragraph starting from row 490.

Point 3: The statistical tests and presents can be improved. The example can be the missing significance test for CGRP expression between different tumor stages. The differences need to be present with statistical calculation.

Response 3: We have added detailed statistical tests in both subsections, the one about serum CGRP and the one with comparation between FSS CGRP and CRLR. For a better presentation of the data, we have modified several tables, especially in the part of comparisons between CGRP and CRLR, where I divided them between low FSS CGRP level and high FSS CGRP level (for better accuracy). We also have added survival curves, to show the prognosis.

At the recommendation of another reviewer, we removed most of the figures from the article, as the presented data is already present in tables and text.

For the document with the responses, please see the attachment

Reviewer 2 Report

Comments and Suggestions for Authors

1.     How does the discussion suggest a potential diagnostic and prognostic role for serum CGRP in colorectal cancer?

2.     What is known about the interaction between nerve structures and tumor cells, particularly in relation to CGRP?

3.     What are the proposed mechanisms by which CGRP may influence tumor growth and progression?

4.     How does CGRP contribute to processes such as motility stimulation, migration, and invasion of tumor cells?

5.     What is the potential therapeutic significance of targeting CGRP in cancer treatment, as discussed in the study?

6.     Which other types of cancer have shown increased serum CGRP levels according to the literature review provided?

Comments on the Quality of English Language

Minor editing of english grammer is needed.

Author Response

Response to Reviewer 2 Comments

Thank you for taking the time to review our manuscript.

All changes are highlighted in red

Point 1: How does the discussion suggest a potential diagnostic and prognostic role for serum CGRP in colorectal cancer?

Response 1: We have added paragraphs in the discussion that suggest the potential diagnostic and prognostic role of serum CGRP. The diagnostic role is due to the fact that serum CGRP levels are much higher in patients with colorectal cancer than in patients from the control group, and the prognostic role is due to the fact that increased levels of CGRP are associated with more advanced stages of the disease and lower survival rate.

Point 2: What is known about the interaction between nerve structures and tumor cells, particularly in relation to CGRP?

Response 2: The interaction between tumor cells and nerve fibers supports the progression of the disease, as demonstrated in pancreatic and prostate cancer, where tumor cells that are near the nerves have less apoptosis and a higher expression of Ki-67 in comparison with distant tumor cells from the nerve cells. Also, certain cells have neuropeptide receptors on their surface, increasing communication between nerves and tumor cells, as demonstrated in esophageal cancer, when inhibition of neuropeptide receptors decreased cell proliferation. Nerve fibers communicate with tumor cells through neuropeptides, such as CGRP or NGF (nerve growth factor), through which they achieve important processes in tumor progression such as angiogenesis and metastasis.

Point 3: What are the proposed mechanisms by which CGRP may influence tumor growth and progression?

Response 3: The proposed mechanisms by which CGRP influence tumor growth and progression are angiogenesis, lymphangiogenesis, perineural invasion and modulating the inflammation.

Point 4: How does CGRP contribute to processes such as motility stimulation, migration, and invasion of tumor cells?

Response 4: CGRP contribute to tumor cells motility stimulation, migration and invasion through the mitogen-activated-protein-kinase (MAPK) signaling pathways.

Point 5: What is the potential therapeutic significance of targeting CGRP in cancer treatment, as discussed in the study?

Response 5: Targeting CGRP in cancer treatment may slow down or even stop the progress of the tumor process and the spread of the disease, (by decreasing angiogenesis, lymphangiogenesis and perineural invasion) considering the increased levels of CGRP that are associated with advanced stages and low survival rates in patients with colorectal cancer.

Point 6: Which other types of cancer have shown increased serum CGRP levels according to the literature review provided?

Response 6: As it is written in the discussions section, serum CGRP have increased levels in other types of cancer, such as prostate, pancreatic, thyroid, breast cancer.

For the document with the responses, please see the attachment

Reviewer 3 Report

Comments and Suggestions for Authors

The manuscript entitled "Expression of Calcitonin Gene-Related Peptide and Calcitonin Receptor-Like Receptor in Colorectal Adenocarcinoma" reports data about serum concentration of CGRP and CRLR as well as histopathological (HP) and immunohistochemical (IHC) analysis in patients with CRC. The study was well conducted, but presentation could be improved. Several data comparisons are repeated in Figures and Tables, specially considering data in Table 2. I suggest to suppress Table 1, since it is not discussed in the text. In the same line, Figures 1 and 2 are not necessary.

Figures 5 and 7 contain the same information available in Table 2. Tables 3-6 could be summarized in 1 graphical figure containing the most relevant information.

In the discussion, the authors do not explore the relation of cause or consequence of these molecules correlation with CRC. In the case of serum diagnostic, this could be considered specific for CRC or any type of cancer or another disease? How specific for CRC is this analysis of CGRP and CRLR?

Comments on the Quality of English Language

Some typos mistakes need to be verified, such as abstract line 42: leves - levels.

Author Response

Response to Reviewer 3 Comments

Thank you for taking the time to review our manuscript.

All changes are highlighted in red

The manuscript entitled "Expression of Calcitonin Gene-Related Peptide and Calcitonin Receptor-Like Receptor in Colorectal Adenocarcinoma" reports data about serum concentration of CGRP and CRLR as well as histopathological (HP) and immunohistochemical (IHC) analysis in patients with CRC. The study was well conducted, but presentation could be improved.

Point 1: Several data comparisons are repeated in Figures and Tables, specially considering data in Table 2. I suggest to suppress Table 1, since it is not discussed in the text. In the same line, Figures 1 and 2 are not necessary.

Response 1: We removed Table 1 and figures 1 and 2.

Point 2: Figures 5 and 7 contain the same information available in Table 2. Tables 3-6 could be summarized in 1 graphical figure containing the most relevant information.

Response 2: We also removed figures 5 and 7 and we created another table with all the information about serum CGRP.

Point 3: In the discussion, the authors do not explore the relation of cause or consequence of these molecules correlation with CRC. In the case of serum diagnostic, this could be considered specific for CRC or any type of cancer or another disease? How specific for CRC is this analysis of CGRP and CRLR?

Response 3: For the correlation part between CGRP and CRLR, I removed tables 3-6 and added 2  new tables, 3 and 4, to show more clearly but at the same time more simply the differences between patients with low FSS and those with high FSS. We have also added graphs about survival in patients with both low FSS and those with high FSS.

We added to the discussion, starting with row 490, the causal relationships between CGRP, CRLR and colorectal cancer.

We also added phrases about the specificity of CGRP and CRLR in colorectal cancer (row 526).

For the document with the responses, please see the attachment

Round 2

Reviewer 1 Report

Comments and Suggestions for Authors

The responses to the questions works fine in most parts. So the important novel part is sample data as well as the study on CGRP and CRLR with CRC stages. 

Double check and modification on biological conclusive description and interpretation would be better such as for correlation between CGRP and CRLR in line 416~418. They positively related then what's the biological meaning? Especially the last paragraph of most results chapters. 

And please double check for the minor typo error like missing "." at line 355.

Author Response

Thank you for taking the time to review our manuscript.

All changes are highlighted in red

The responses to the questions works fine in most parts. So the important novel part is sample data as well as the study on CGRP and CRLR with CRC stages.

Double check and modification on biological conclusive description and interpretation would be better such as for correlation between CGRP and CRLR in line 416~418. They positively related then what's the biological meaning? Especially the last paragraph of most results chapters.

Response 1: We added a phrase with the biological meaning of the correlation between CGRP and CRLR from line 416, and also concluding paragraphs with a biological description at the end of each subsection of the results chapter.

Point 2: And please double check for the minor typo error like missing "." at line 355.

Response 2:  We checked and modified all typing errors in the manuscript